# Phytoglobin Expression Alters the Na^+^/K^+^ Balance and Antioxidant Responses in Soybean Plants Exposed to Na_2_SO_4_

**DOI:** 10.3390/ijms23084072

**Published:** 2022-04-07

**Authors:** Mohamed S. Youssef, Mohammed M. Mira, Sylvie Renault, Robert D. Hill, Claudio Stasolla

**Affiliations:** 1Department of Plant Science, University of Manitoba, Winnipeg, MB R3T 2N2, Canada; mohamedsamir.youssef@umanitoba.ca (M.S.Y.); mohammed.mira@umanitoba.ca (M.M.M.); rob.hill@umanitoba.ca (R.D.H.); 2Botany and Microbiology Department, Faculty of Science, Kafrelsheikh University, Kafrelsheikh 33516, Egypt; 3Department of Botany and Microbiology, Faculty of Science, Tanta University, Tanta 31527, Egypt; 4Department of Biological Sciences, University of Manitoba, Winnipeg, MB R3T 2N2, Canada; sylvie.renault@umanitoba.ca

**Keywords:** antioxidant response, phytoglobin, reactive oxygen species, sodium sulfate, soybean, salinity tolerance, Na^+^/K^+^ balance

## Abstract

Soybean (*Glycine max*) is an economically important crop which is very susceptible to salt stress. Tolerance to Na_2_SO_4_ stress was evaluated in soybean plants overexpressing or suppressing the phytoglobin *GmPgb1*. Salt stress depressed several gas exchange parameters, including the photosynthetic rate, caused leaf damage, and reduced the water content and dry weights. Lower expression of respiratory burst oxidase homologs (*RBOHB* and *D*), as well as enhanced antioxidant activity, resulting from *GmPgb1* overexpression, limited ROS-induced damage in salt-stressed leaf tissue. The leaves also exhibited higher activities of the H_2_O_2_-quenching enzymes, catalase (CAT) and ascorbate peroxidase (APX), as well as enhanced levels of ascorbic acid. Relative to WT and *GmPgb1*-suppressing plants, overexpression of *GmPgb1* attenuated the accumulation of foliar Na^+^ and exhibited a lower Na^+^/K^+^ ratio. These changes were attributed to the induction of the Na^+^ efflux transporter SALT OVERLY SENSITIVE 1 (SOS1) limiting Na^+^ intake and transport and the inward rectifying K^+^ channel POTASSIUM TRANSPORTER 1 (AKT1) required for the maintenance of the Na^+^/K^+^ balance.

## 1. Introduction

Salinity influences plant growth and development and has negative effects on productivity [1]. Occurring virtually under all climatic conditions as a result of natural events and human practices, salinity is more prominent in arid environments where the discrepancy between the plant demand for water and the water provided by precipitation results in the accumulation of minerals in the soil [2]. It is estimated that about 20% of irrigated land is salt-affected [3], and plant survival is dependent on the salinity tolerance of the species. Soybean (*Glycine max*), a valuable source of proteins and oils, is especially susceptible to high salt concentrations, and several desirable agronomic traits during both vegetative and reproductive stages are severely affected [4]. Among the different stages of development, seedlings are the most vulnerable to salt stress, as demonstrated by substantial growth retardation which compromises the subsequent phases of development and, ultimately, plant productivity [5].

Plants exposed to salinity experience several types of stress including osmotic, ionic, and oxidative stress. Osmotic stress arises from a reduction of the soil’s osmotic potential, which impairs the ability of plants to absorb water [6]. Ionic stress occurs as a result of toxic accumulation of ions, especially of Na^+^, which contributes to mineral imbalance and nutrient deficiency, as well as damage of cellular components [7]. Oxidative stress is caused by excessive accumulation of reactive oxygen species (ROS) causing cellular damage [8].

The combination of these effects leads to two temporally separated responses. The initial inhibition of growth, occurring within hours as a result of depressed cell elongation and decreased stomatal conductance, is followed by damage resulting from nutritional imbalance and oxidative stress [9,10]. Therefore, mitigation of nutritional imbalance through maintenance of ion homeostasis and attenuation of oxidative stress through the activation of antioxidant responses are desirable traits in salinity tolerance [11], also reported in soybean [5].

The main factor contributing to salinity stress is Na^+^, which alters water management, damages cellular organelles and membranes, and impedes energetic and photosynthetic processes [12]. Besides depressing the activity of many enzymes involved in the synthesis of chlorophyll, Na^+^ impairs components of the photosynthetic electron transport chain [13,14] and proteins required for the reduction of CO_2_ [15]. Many of these effects have been linked to the Na^+^ disturbance in the cellular level of K^+^ which is required in many photosynthetic functions [16]. Noxious levels of Na^+^, in fact, not only competitively inhibit the K^+^ influx, but also induce the K^+^ efflux through depolarization of plasma membranes [17]. Therefore, maintenance of a low Na^+^/K^+^ ratio through the retention of cellular K^+^ and/or exclusion of Na^+^ from sensitive tissues is a desirable trait of salt tolerance [18]. This notion is best exemplified by the acquired tolerance to salinity resulting from supplementation of K^+^ or exclusion of cellular Na^+^ through the upregulation of Na^+^ efflux transporters [19,20,21,22].

Besides altering the K^+^ homeostasis, excess Na^+^ causes oxidative stress through accumulation of reactive oxygen species (ROS) [8]. A soybean Na^+^/H^+^ antiporter, SALT-OVERLY SENSITIVE 1 (SOS1), alleviated toxicity by limiting the Na^+^ accumulation in leaves [22]. Overexpression of *SOS1* also mitigated oxidative stress resulting from ROS accumulation [22]. Required at low levels to regulate many aspects of growth and development, ROS can become noxious if they exceed physiological thresholds, as observed during different conditions of stress, including salinity [23]. Elevated levels of ROS alter the function of membranes by oxidizing fatty acids and lipids and cause inhibition of enzymes and damage to nucleic acids. If not attenuated, these effects culminate in cell death [24,25], as also reported in soybean plants grown under saline conditions [5].

Cellular ROS homeostasis is modulated by their synthesis, mainly through the NADPH oxidase complex, and their scavenging through the antioxidant machinery with glutathione and ascorbate acting as major antioxidants [26]. Soybean plants exposed to salt stress exhibit a rapid increase in the activities of both superoxide dismutase (SOD) and catalase (CAT) [22] and major changes in the activity of the enzymes participating in the ascorbate–glutathione redox system [5]. An enhanced antioxidant system in response to salinity is, thus, a major component in dealing with stress [5].

Among the modulators of ROS and antioxidant responses is nitric oxide (NO) [27]. Overproduction of NO during several stress conditions such as hypoxia or water deficit augments ROS and depresses the activities of several antioxidant enzymes, thus triggering cellular and tissue damage which compromise plant survival [28,29,30]. Nitric oxide also modulates the Na^+^/K^+^ homeostasis during salinity stress through regulation of the inward rectifying K^+^ channel POTASSIUM TRANSPORTER 1 (AKT1), albeit with contrasting results [31,32]. In these studies, cellular NO levels were modulated using pharmacological applications, i.e., NO donors or scavengers, which pose some challenges when interpreting the results due to uniformity of applications and duration of the effect of treatment [33]. The same authors acknowledged that the mechanisms of NO action in salinity stress could be better analyzed using a transgenic approach.

Modulating NO can be achieved using transgenic plants with different levels of phytoglobins (Pgbs); plant proteins known for their NO scavenging properties [34]. Induced at the onset of abiotic and biotic stress, Pgbs have been shown to play a protective role by attenuating oxidative stress through cell- and tissue-specific suppression of NO [28,30,35,36]. Overexpression of the soybean *GmPgb1* mitigated the growth inhibition of waterlogging [37] and drought [38] through reduction of NO, ROS, and oxidative stress. To further our knowledge on the function of Pgb1 as a stress attenuator, we examined the behavior of previously characterized soybean plants overexpressing or downregulating *GmPgb1* [37] to Na_2_SO_4_ stress. It is hypothesized that during salt stress, overexpression of *GmPgb1* attenuates the depression of several morphophysiological parameters through limitation of ROS-induced oxidative damage and Na^+^ accumulation. To address this hypothesis, we measured biomass, water content, leaf damage, and K^+^ and Na^+^ concentrations, as well as photosynthetic gas exchange parameters and the enzymes related to oxidative stress known to be influenced by salinity.

## 2. Results

### 2.1. Overexpression of GmPgb1 Augments Tolerance to Salinity Stress

The transcript levels of *GmPgb1* during Na_2_SO_4_ stress were measured in the WT line and in previously characterized lines [37] overexpressing (GmPgb1(S)17) or downregulating (GmPgb1(RNAi)23) *GmPgb1*. Salinity gradually augmented the *GmPgb1* transcripts in WT roots and leaves, with a marked increment observed in the latter between 6 and 12 h (Figure 1). The increase in *GmPgb1* in leaves trailed the increase in roots by approximately 6 h. The *GmPgb1* overexpression resulted in an approximately six-fold increase in root transcripts and an approximately twelve-fold increase in leaf transcripts, likely reflecting the relatively low base levels of transcripts in leaves compared to roots of non-transformed plant materials [38]. The RNAi line did not completely repress the induction of *GmPgb1* in roots whereas it did repress the leaf *GmPgb1* expression.

Differences in photosynthetic parameters between the lines were clearly detectable after 24 h of Na_2_SO_4_ treatment. In leaves of WT plants, salinity stress depressed the photosynthetic rate, the transpiration rate, the stomatal conductance, and the water use efficiency by more than 50% (Figure 2). With the exclusion of transpiration, leaves overexpressing *GmPgb1* exhibited a higher retention of gas exchange parameters. The GmPgb1(RNAi)23 line had the lowest retention of the photosynthetic rate, the transpiration, and the stomatal conductance relative to the WT (Figure 2). Salinity also induced the closure of the stomata (Appendix A), as evidenced by the reduced stomatal conductance. This effect was mitigated in leaves of the GmPgb1(S)17 line. During stress, reduction in the photosynthetic rate is often concomitant to the degradation of thylakoid-associated protein complexes [39]. The transcript levels of the selected genes known to be sensitive to abiotic stress and involved in the formation of photosystem I (PsaA, PsaB, and LHCA), cytochrome B6F (cylB6F), and photosystem II (PsbA, PsbB, PsbC, and PsbD) complexes were measured at 24 h. Except for LHCA, the relative transcript levels of all the genes, comparable between genotypes under control conditions, decreased in leaves of the WT plants exposed to Na_2_SO_4_ (Appendix A). This decrease was mitigated by the overexpression of *GmPgb1*. Relative to the WT, GmPgb1(RNAi)23 leaves exhibited lower levels of PsaA, PsbB, and PsbC (Appendix A).

Morphological differences between the lines were visible after seven days of stress. Overexpression of *GmPgb1* limited the Na_2_SO_4_ depression of the shoot water content (WC) and the root and shoot dry weight, which were more severe in the WT and GmPgb1(RNAi)23 lines (Figure 3). Leaf injury, not noticeable in leaves of the GmPgb1(S)17 line exposed to salinity for seven days, was apparent in the other two lines, and especially in leaves of the GmPgb1(RNAi)23 plants (Figure 3).

### 2.2. Na^+^ and K^+^ Accumulation during Salinity Stress Is Influenced by GmPgb1

Depression of the photosynthetic rate and leaf injury during salinity stress are often related to the toxicity of the Na^+^ [13,14] accumulating in leaves and disturbing the homeostasis of K^+^ needed in several photosynthetic functions [16]. Salt stress augmented root and leaf Na^+^ in all the genotypes, but the increment in leaves was reduced in plants overexpressing *GmPgb1* (Figure 4A). The level of K^+^ was generally reduced by Na_2_SO_4_ in both organs, and the reduction was abolished when *GmPgb1* was overexpressed (Figure 4B). Because of these changes, the foliar Na^+^/K^+^ ratio after 24 h of salt stress was lower in the leaves overexpressing *GmPgb1* relative to those of WT and GmPgb1(RNAi)23 plants (Figure 4C)

To further understand the mechanisms for the differential accumulation of Na^+^ and K^+^, we measured the expression of two key genes involved in Na^+^ and K^+^ transport: the Na^+^ efflux transporter SALT-OVERLY SENSITIVE 1 (SOS1) limiting the Na^+^ intake in roots and its long-distance movement [16] and the inward-rectifying K^+^ channel AKT1 required for the maintenance of the Na^+^/K^+^ ratio [40]. The expression of *SOS1*, which remained unchanged in GmPgb1(RNAi)23 roots and leaves following imposition of Na_2_SO_4_ stress, increased in both organs of the plants overexpressing *GmPgb1* (Figure 4D). A small significant increase also occurred in WT leaves. Salinity also elevated the expression of *AKT1* in the roots of GmPgb1(S)17 plants.

### 2.3. Na_2_SO_4_ Accumulation of Foliar H_2_O_2_ and Lipid Peroxidation Are Mitigated by the Overexpression of GmPgb1

A consequence of Na^+^ toxicity is the overproduction of ROS causing severe cellular damage including peroxidation of lipid membranes, which can be estimated by the level of MDA [41]. In both the WT and GmPgb1(RNAi)23 leaves, exposure to salinity for 24 h elevated foliar H_2_O_2_ and MDA (Figure 5A). Salinity increased both H_2_O_2_ and MDA, whereas in the stressed plants overexpressing *GmPgb1*, the increase in H_2_O_2_ was limited, and the level of MDA was not different from the control (Figure 5).

A similar pattern was observed for the expression of two respiratory burst oxidase homologs (RBOHB and D), components of the ROS generator NADPH oxidase enzyme [42] and indicators of ROS production in soybean [37]. The increase in foliar expression of both genes, more apparent after 6 h of treatment, was in fact attenuated by the overexpression of *GmPgb1* (Figure 5B). No differences between the genotypes were observed for *RBOHG*.

### 2.4. Antioxidant Responses in Leaves of Salt-Stressed Plants Are Induced by GmPgb1

Limitation of oxidative damage is a tolerance trait in several forms of abiotic stress, including salinity [43]. The cellular level of ROS is determined not only by their production, but also by the antioxidant system composed by an array of enzymes. These include superoxide dismutase (SOD), scavenging superoxide; catalases (CAT) and ascorbate peroxidase (APX) participating in the detoxification of H_2_O_2_; and enzymes required for the turnover of ascorbic acid, dehydroascorbate reductase (DHAR), monodehydroascorbate reductase (MDHAR), and glutathione reductase (GR).

Expression analyses of the genes encoding these enzymes and responsive to abiotic stress [37] were performed. The foliar expression of *SOD1* and *CAT1* increased after 6 h of Na_2_SO_4_ stress (Figure 6A). This increase was more pronounced in the GmPgb1(S)17 plants and lower in the the GmPgb1(RNAi)23 line relative to the WT. A similar profile was also observed for *CAT2* despite an earlier (0–6 h) increase in expression. Salinity induced *SOD3* in the three genotypes, and after 24 h, the expression of this gene was greatest in the GmPgb1(S)17 plants, intermediate in the WT plants, and lowest in those suppressing *GmPgb1* (Figure 6A).

Transcript levels of the enzymes participating in glutathione and ascorbate metabolism were also measured. Salinity increased the foliar expression of *APX1*, especially in the GmPgb1(S)17 line (Figure 6B). No major differences between the lines were observed for *MDHAR1* and 4 despite an increase in expression in the WT leaves after 12 h. The expression of *DHAR1* increased between 0 and 6 h independently of the genotype; at 24 h, a higher expression level was observed in leaves, upregulating *GmPgb1*. Of the two *GR* genes, the expression of *GR1* did not exhibit pronounced fluctuations during the Na_2_SO_4_ treatment, while a gradual suppression of *GR2* was recorded in the GmPgb1(S)17 line. This contrasted with the WT and the GmPgb1(RNAi)23 line which showed an increase in *GR2* transcripts after 6 h (Figure 6B).

To further interpret the changes in transcript levels, the activities of the antioxidant enzymes were measured in leaves of plants grown for 24 h under control or Na_2_SO_4_ conditions. The activities of many enzymes, similar under control conditions among the three genotypes, were induced by salt stress (Figure 7). Relative to the WT and GmPgb1(S)23 lines, the foliar activities of APX, DHAR, and CAT were higher in the GmPgb1(S)17 line under salinity stress. The three lines exhibited similar activity values for MDHAR and GR, while the activity of SOD was slightly reduced in the GmPgb1(RNAi)23 line when compared to the values of the other two lines exposed to stress (Figure 7).

### 2.5. GmPgb1 Expression in Commercial Soybean Cultivars

The increased salinity tolerance in plants overexpressing *GmPgb1* prompted an analysis of the natural germplasm to determine if a relationship exists between *GmPgb1* expression and salt tolerance. Based on the combined analyses of several morphological parameters used on 18 commercial cultivars (Appendix A), we measured the level of *GmPgb1* in three cultivars (1–3) exhibiting the highest tolerance to Na_2_SO_4_ and three cultivars (16–18) exhibiting the highest susceptibility (Figure 8). The expression of *GmPgb1* increased during the first 12 h of stress in roots and especially shoots of tolerant lines while it remained low in the susceptible cultivars (Figure 8).

## 3. Discussion

Overexpression of *GmPgb1* alleviated salinity stress in soybean by mitigating the salt-induced damage and the reduction of several morphophysiological parameters. In both WT and *GmPgb1*-downregulating plants, exposure to Na_2_SO_4_ caused a reduction in biomass and a depression of the photosynthetic rate, the stomatal conductance, the transpiration, and the water use efficiency (Figure 2). These effects, typical of salt-treated soybean plants [5,6], may be the consequence of the combinatorial effects of salinity resulting from the osmotic stress reducing the plant’s hydraulic conductivity and, most importantly, ionic stress caused by the toxic accumulation of ions, especially of Na^+^ [6,7,22,44]. Noxious levels of Na^+^ depress photosynthesis [13] by disrupting the function of the photosystem II reaction center and the oxygen-evolving complex [3]. In soybean, this can be further aggravated by the concomitant reduction in foliar K^+^ (Figure 4) attributable to the Na^+^ interference with its translocation [17,45]. Many photosynthetic reactions, in fact, require K^+^ [16].

Salt-treated WT plants also experience oxidative stress, as evidenced by the increase in H_2_O_2_ and MDA, a marker of lipid peroxidation (Figure 5A), and these effects can be linked to excessive levels of Na^+^ [46,47]. Consistent with this observation is the transcriptional induction of *RBOHs* (Figure 5B), components of the NADPH oxidase multicomplex system [41] and markers of ROS production [37], and the activation of several ROS-detoxifying enzymes, including APX, CAT, and SOD (Figure 5A). Thus, the responses observed in Na_2_SO_4_ in the WT leaves are typical in many respects to those observed in many species [46], including soybean [5].

Overexpression of *GmPgb1* ameliorates plant response to salinity through the retention of photosynthetic gas exchange parameters and mitigates foliar tissue damage (Figure 2 and Figure 3). These effects are associated to the ability of these plants to limit accumulation of Na^+^ (and retain a low Na^+^/K^+^ ratio) and mitigate oxidative stress. A plausible reason for the reduced Na^+^ levels in tissue overexpressing *GmPgb1* is the increase in *SOS1* (Figure 4D), a member of the SOS pathway contributing to salinity tolerance [48]. Strategically located in epidermal, cortical, and pericycle cells, SOS1 limits the Na^+^ intake in root cells, restricts the radial movement of the ion across cells layers, thereby reducing its long-distance transport through the xylem, and excludes Na^+^ in leaf cells [16]. The induction of *GmSOS1* in both roots and leaves of the Na_2_SO_4_-treated GmPgb1(S)17 plants, consistent with the higher expression levels observed in species tolerant to salt stress [49] and the acquired tolerance of transgenic plants overexpressing SOS1 [20], suggests limited Na^+^ intake in root cells and transport to the aerial organs.

The increased expression of *GmATK1* in the roots of the salt stress-exposed GmPgb1(S)17 plants (Figure 4D) is also an indication of increased tolerance [45,50]. As an inward-rectifying K^+^ channel, ATK1 is required for the maintenance of the Na^+^/K^+^ balance by ensuring sustained K^+^ intake, especially in high salt environments. The crucial function of this transporter during salinity stress was demonstrated by Wang et al. [45] documenting enhanced tolerance and retention of the Na^+^/K^+^ balance in *Arabidopsis* plants overexpressing the soybean *GmATK1*.

Besides restricting Na^+^ from leaves, overexpression of *GmPgb1* alleviates salt damage by relieving oxidative stress, specifically reducing the level of ROS and ROS-induced damage. This protective role, documented under developmental and stress events [26,29,30], is linked to the function of the protein to scavenge NO [34], as also observed for GmPgb1 (Appendix A). Here, we show that expression of *GmPgb1* during salt stress attenuates the increase in H_2_O_2_ and MDA, possibly by limiting the Na_2_SO_4_ induction of *RBOHB* and *RBOHD*, markers of ROS production in soybean [37]. The mitigation of ROS production in *GmPgb1*-overexpressing leaves is a contributing factor for a decreased leaf injury and a higher retention of the photosynthetic rate. Additionally, overexpression of *GmPgb1* also attenuates the Na_2_SO_4_ depression of the transcripts involved in the formation of photosystem I (PsaA and B), cytochrome B6F (cylB6F), and photosystem II (PsbB, PsbC, PsbD). Retention of these transcripts during conditions of stress has been associated to abiotic stress tolerance [37].

Reduced oxidative stress in leaves overexpressing *GmPgb1* can be attributed to an improved antioxidant system. Relative to the WT, overexpression of *GmPgb1* is associated to higher levels of the *CAT1*, *2* and *APX1* transcripts, as well as the CAT and APX activity. This behavior might be accountable for the lower level of H_2_O_2_ measured in the GmPgb1(S)17 leaves during salt stress. Elevated activity of CAT and APX are desirable tolerance traits for salinity stress [22]. While the direct mechanisms through which *GmPgb1* modulates the expression and activity of *CAT* and APX1 are unknown, regulation of the APX1 activity might be the result of post-translational modifications mediated by NO. Nitration of APX by peroxynitrate, formed in environments enriched in NO, inhibits the enzyme [51], an observation consistent with the higher APX activity (Figure 7) in *GmPgb1* leaves which are depleted in NO (Appendix A).

Sustained activity of APX and removal of H_2_O_2_ during conditions of stress is dependent upon the availability of the substrate ascorbate produced de novo and regenerated through the ascorbate–glutathione cycle [43]. Our work suggests that during salt stress, overexpression of *GmPgb1* might result in higher levels of ascorbate required for the operation of APX. First, the expression of the three ascorbate biosynthetic genes GDP-L-galactose phosphorylase GmGPP and L-galactose-dehydrogenase (GmGDH1 and 2), good indicators of ascorbate production in soybean [37,52], is induced in leaves of Na_2_SO_4_-treated plants overexpressing *GmPgb1* (Appendix A). Second, the same leaves exhibit a higher activity of DHAR, which is required for ascorbate recycling through the glutathione-dependent dehydroascorbate reduction [23]. The relevance of this enzyme in replenishing the ascorbate pool during salinity stress was documented [53]. The slightly lower ascorbate level in the salt-stressed GmPgb1(S)17 leaves (Appendix A) is difficult to interpret, but it might be a consequence of higher utilization by APX. While we suggest that removal of ROS by ascorbate is mediated by APX, the operation of APX-independent mechanisms, which also involve ascorbate [54], cannot be discounted. Activation of the antioxidant components and similar Na^+^/K^+^ metabolic adaptations to those described here were associated to the ability of thymol to increase salinity tolerance in tobacco plants [15]. In the same study, it was observed that thymol elevates the level of NO, an observation which is, however, not consistent with our study.

The relevance of high expression of *GmPgb1* as a mechanism to cope with salinity stress is not limited to transgenic material. Our study shows that this gene is preferentially induced in commercial cultivars exhibiting tolerance to Na_2_SO_4_, relative to those susceptible to the same stress. This observation, consistent with a previous work linking induction of *Pgbs* to the ability of soybean plants to cope with hypoxic stress [37], could be exploited as a tool to screen plant material at early stages of development for salt stress tolerance.

In conclusion, this study demonstrates that overexpression of the soybean *GmPgb1* enhances tolerance to salt stress. The Na_2_SO_4_ depression of gas exchange parameters, water use efficiency, leaf damage, and reduction in water content are attenuated by the overexpression of *GmPgb1*. These beneficial effects can be attributed to the ability of plants overexpressing *GmPgb1* to limit the accumulation of Na^+^ in leaf tissue (thus maintaining a lower Na^+^/K^+^ ratio) and mitigate oxidative stress. This is achieved by reducing production of ROS as well as by activating antioxidant responses centered around the activities of CAT and APX.

Collectively, these data suggest GmPgb1 as an effective modulator of soybean response to salt stress.

## 4. Materials and Methods

### 4.1. Plant Material and Growth Conditions

Transformed soybean (*Glycine max*) plants with altered levels of the *Phytoglobin1* gene *GmPgb1*, described by Mira et al. [37], were selected for this study. Soybean cultivars grown in Western Canada were provided by Dennis Lange (Manitoba Agriculture, Winnipeg, Canada). These cultivars were the same as the ones utilized previously [55].

Seed sterilization was carried out with chlorine gas for 20 h [56]. The sterilized seeds were germinated on moistened filter paper for about one week. Uniform seedlings characterized by unfolded unifoliate leaves (VC stage of development) were selected and grown further hydroponically in a solution containing half-strength Hoagland [57]. Acclimation to the hydroponic conditions was conducted for 7 days under a 16 h photoperiod of 22 °C (light) and 18 °C (dark), with 334 µmol m^−2^ s^−1^ light intensity. As soon as the plants developed their first trifoliate leaf (V1 stage of development) they were transferred to a half-strength Hoagland solution containing 100 mM Na_2_SO_4_. The level of Na_2_SO_4_ and the timing of the measurements were determined empirically to ensure consistent manifestation of symptoms. Plants in the control group were given identical treatments as the experimental plants, except for Na_2_SO_4_. The plants were harvested at different timepoints for morphophysiological parameter assessments, as indicated below. Unless otherwise specified, all the experiments carried out used at least five biological replicates, each with a minimum of five plants.

### 4.2. Determination of Morphophysiological Parameters

Fresh and oven-dry weights of shoot or root tissues were measured after 7 days of treatment. The extent of leaf injury, also measured at 7 days, was assessed using a visual scale adopted in a previous work [55], based on an index (0–100) reflecting the percentage of foliar damage.

Gas exchange parameters (photosynthetic rate, stomatal conductivity, water use efficiency, and transpiration rate) were measured on the first trifoliate leaves after 24 h of treatment using a gas exchange infrared gas analyzer (IRGA; LI-6400, LI-COR, Inc., Lincoln, NE, USA), as reported by Youssef et al. [55]. The measurements were conducted between 9 am and 3 pm. Gas exchange parameters were measured under photosynthetically active radiation of 400 μmol m^−2^ s^−1^, atmospheric CO_2_ of 400 μmol mol^−1^, relative humidity of 50%, and temperature of 22 °C. The measurements were performed on three biological replicates, each consisting of at least five plants.

### 4.3. Measurements of K^+^ and Na^+^

Leaf and root tissues (2 g) were harvested at 24 h under control or Na_2_SO_4_ treatment and oven-dried for 5 days. Analysis of K^+^ and Na^+^ was performed using Agvise (Agvise Laboratories Inc., Northwood, ND, USA; www.agvise.com, 29 March 2022). Extraction of both elements was performed using 1 M ammonium acetate (pH 7), and determination was carried out by means of inductively coupled plasma atomic emission spectroscopy (ICP-AES).

### 4.4. Stomata Measurements

Analysis of stomata was performed after 24 of stress [55]. Briefly, a thin layer of translucent nail polish was applied to a leaf’s abaxial side and left to dry for 5 min. A transparent tape was applied on the dry nail polish and carefully peeled off before being placed on a clean microscope slide for microscopic examination. Stomata were categorized into three groups based on their length-to-width ratio (L/W): fully open (L/W = 1–2), partially open (L/W = 2.1–2.5), or closed (L/W 2.6) (Appendix A). The analyses were conducted using three biological replicates each, with at least 100 stomata.

### 4.5. Measurements of Transcript Levels

Transcript levels were measured on the first trifoliate leaves at 0, 6, 12, and 24 h of stress. Tissue was harvested and stored at −80 °C. Total RNA was extracted with a TRI Reagent solution (Invitrogen, Waltham, MA, USA) following the manufacturer’s instructions. The RNA samples were treated with DNase (Thermo Fisher Scientific, Waltham, MA, USA), and a High Capacity cDNA Reverse Transcription Kit (Applied Biosystems, Waltham, MA, USA) was used for cDNA synthesis. The relative levels of transcripts were measured by means of quantitative PCR using the primers listed in Appendix A and analyzed with the 2^−ΔΔCT^ method [58] using helicase (Gm04g07180.4) as the reference gene.

### 4.6. Antioxidant Enzyme Activity and Quantification of Ascorbic Acid and Glutathione

The enzyme activity of APX, DHAR, MDHAR, and GR were measured as reported by Zhang and Kirkham [59]. Trifoliate leaf tissue (100–150 mg fresh weight), collected at 24 h under control or Na_2_SO_4_ treatment, was immediately homogenized on ice with 50 mM phosphate buffer (pH 7.8) containing 0.2 mM EDTA and 1% (*w*/*v*) polyvinylpyrrolidone. The homogenate was then centrifuged at 18,000× *g* at 4 °C for 15 min. The supernatant was collected for the analysis of enzyme activities [59] using a Bio-Tek Synergy H4 Microplate Reader.

The APX activity was estimated by measuring the H_2_O_2_-dependent oxidation of ascorbic acid following the decrease in absorbance at 290 nm. The reaction mixture contained 0.5 mM ASC, 0.1 mM H_2_O_2_, 0.1 mM EDTA, and 50 mM sodium phosphate buffer (pH 7.0).

The DHAR activity was measured by following the GSH-dependent production of ASC at 265 nm. The reaction mixture contained 2.5 mM GSH, 0.1 mM EDTA, 0.2 mM dehydroascorbate (DHA), and 50 mM sodium phosphate buffer (pH 7.0).

The MDHAR activity was measured by observing the change in absorbance at 340 nm. The reaction mixture contained 0.1 mM NADH, 2.5 mM ASC, 50 mM sodium phosphate buffer (pH 7.6), and four units of ASC oxidase.

The GR activity was determined by observing the NADPH-dependent oxidation of GSSG following the decrease in absorbance at 340 nm. The reaction mixture contained 1 mM EDTA, 1 mM GSSG, 0.2 mM NADPH, and 0.1 M sodium phosphate buffer (pH 7.8).

For CAT and SOD, leaf tissue (100–150 mg fresh weight) was homogenized on ice-cold 50 mM sodium phosphate buffer (pH 7.0) containing 0.2 mM EDTA and 1% (*w*/*v*) polyvinylpyrrolidone. The homogenate was centrifuged at 15,000× *g* at 4 °C for 20 min. The supernatant was then collected for the enzyme activity analysis [59].

For the analysis of ascorbic acid (ASC) and dehydroascorbate (DHA), about 100–150 mg (fresh weight) of leaf tissue was first powdered in liquid nitrogen and then homogenized with 2 mL of ice-cold 5% metaphosphoric acid. The homogenate was centrifuged at 21,000× *g* at 4 °C for 15 min. The supernatant was collected and the absorbance (525 nm) was read [59].

### 4.7. Lipid Peroxidation and H_2_O_2_ Measurements

Lipid peroxidation was estimated by measuring the amount of malondialdehyde (MDA) [60]. The leaves (about 0.5 g) collected at 24 h under control or Na_2_SO_4_ treatment were homogenized in 10 mL of 0.1% (*w*/*v*) trichloroacetic acid (TCA), centrifuged at 15,000× *g* for 10 min, and 1 mL of the supernatant was added to 4 mL thiobarbituric acid (TBA) in 20% (*w*/*v*) TCA. The solution was heated at 95 °C for 30 min centrifuged at 10,000× *g* for 10 min and used to measure the absorbance at 532 nm. A standard curve was generated to determine the content of MDA in the tissue.

Hydrogen peroxide was measured in trifoliate leaves using the ferrous oxidation protocol with xylenol orange [61]. The level of H_2_O_2_ was determined using a standard curve.

### 4.8. Nitric Oxide (NO) Localization

The accumulation of intracellular NO in the leaf tissue was monitored using the NO-specific fluorescent probe DAF-FM DA (MilliporeSigma, Burlington, MA, USA), as described previously [62,63]. The leaves, collected at 24 h under control or Na_2_SO_4_ treatment, were incubated for 1 h at 37 °C with gentle agitation in the dark in a buffer (50 mM Tris and 50 mM KCl, pH 7.2) containing 1% (*v*/*v*) Triton X-100 and 50 μM DAF-FM DA. The leaves were rinsed twice in the same buffer (50 mM Tris and 50 mM KCl, pH 7.2). The fluorescent signal was captured with a GFP light cube in a EVOS M5000 Imaging System, with excitation and emission peaks at 470 and 510 nm. Image J was used to determine the DAF-FM DA fluorescent pixel intensity in leaf segments.

### 4.9. Statistical Analysis

The data were analysed by means of one-way analysis of variance (ANOVA) using the SPSS program (version 19.0). Treatment means were compared using Duncan’s test (α = 0.05) to differentiate the significance of the difference between various parameters.

## Figures and Tables

**Figure 1 ijms-23-04072-f001:**
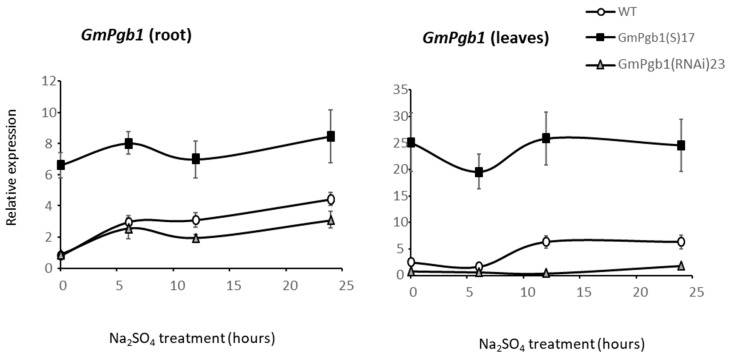
Expression level of *GmPgb1* in roots and leaves of soybean plants exposed to Na_2_SO_4_ for 24 h. Measurements were taken in the WT line and lines overexpressing (GmPgb1(S)17) or downregulating (GmPgb1(RNAi)23) *GmPgb1*. Values are the means + SE of three biological replicates, each consisting of at least three plants.

**Figure 2 ijms-23-04072-f002:**
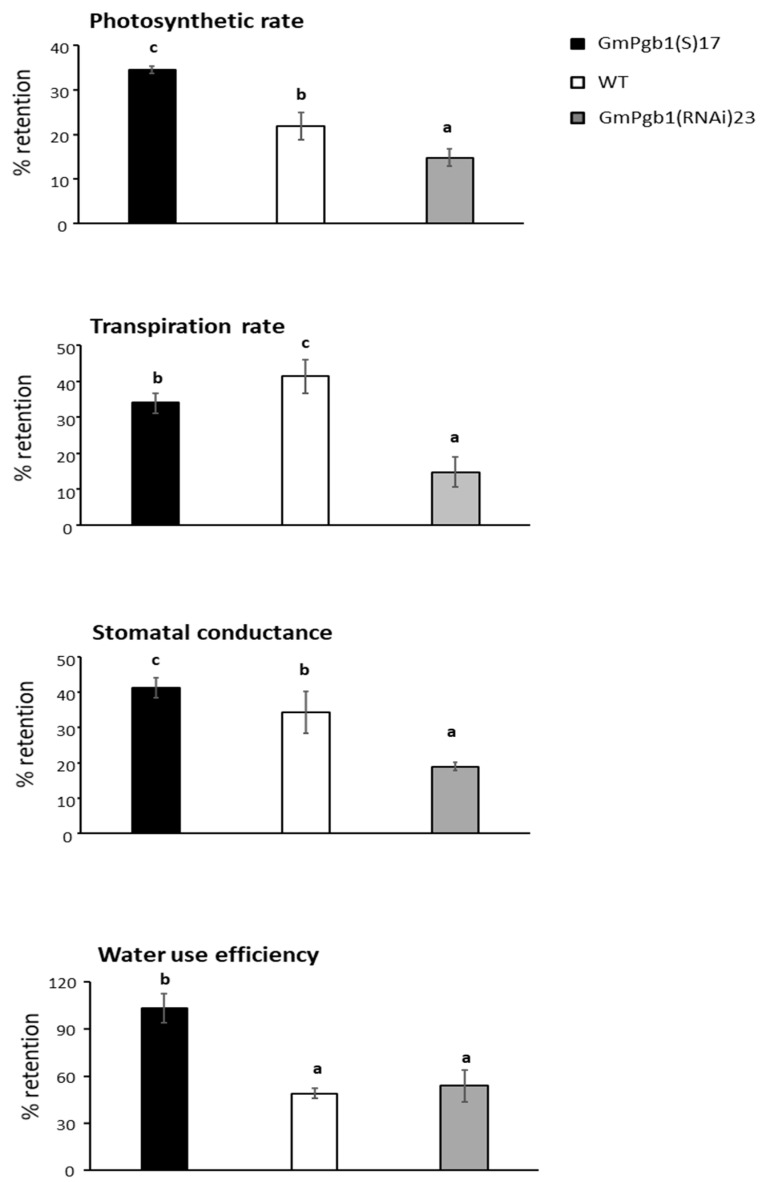
Retention, expressed as the percentage of control (Na_2_SO_4_), of the photosynthetic rate, the transpiration rate, the stomatal conductance, and the water use efficiency of Na_2_SO_4_-treated soybean plants overexpressing (GmPgb1(S)17) or downregulating (GmPgb1(RNAi)23) *GmPgb1*. Measurements were conducted after 24 h of treatment. Values are the means ± SE of at least three biological replicates each consisting of at least five plants. Letters indicate statistically significant differences (*p* < 0.05).

**Figure 3 ijms-23-04072-f003:**
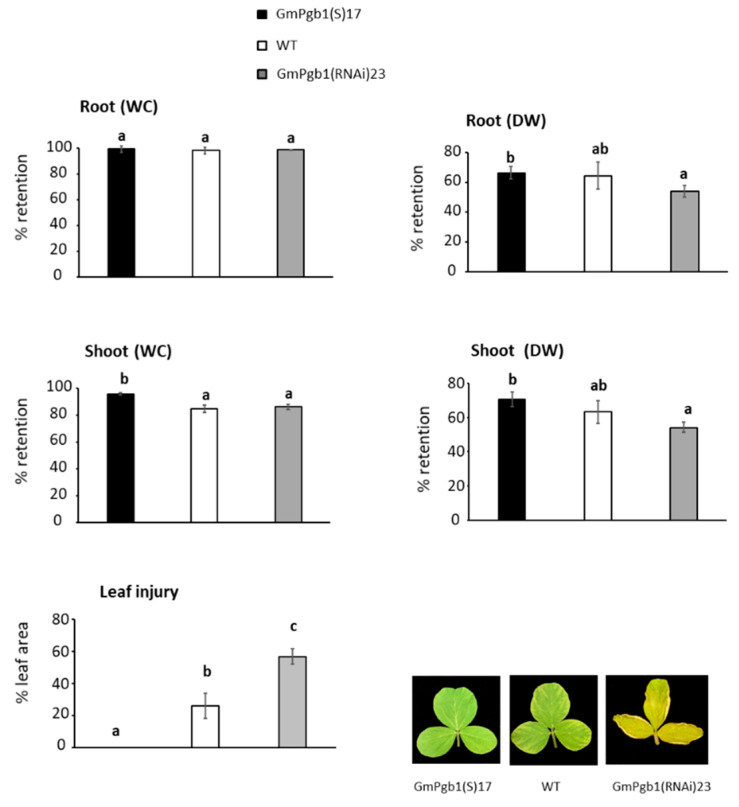
Retention of the water content (WC) and dry weight and leaf injury of soybean plants exposed to Na_2_SO_4_ for seven days. Measurements were taken in the WT line and lines overexpressing (GmPgb1(S)17) or downregulating (GmPgb1(RNAi)23) *GmPgb1*. Values are the means ± SE of five biological replicates, each consisting of at least five plants. Letters indicate statistically significant differences (*p* < 0.05). Micrographs of leaves of waterlogged plants are also shown.

**Figure 4 ijms-23-04072-f004:**
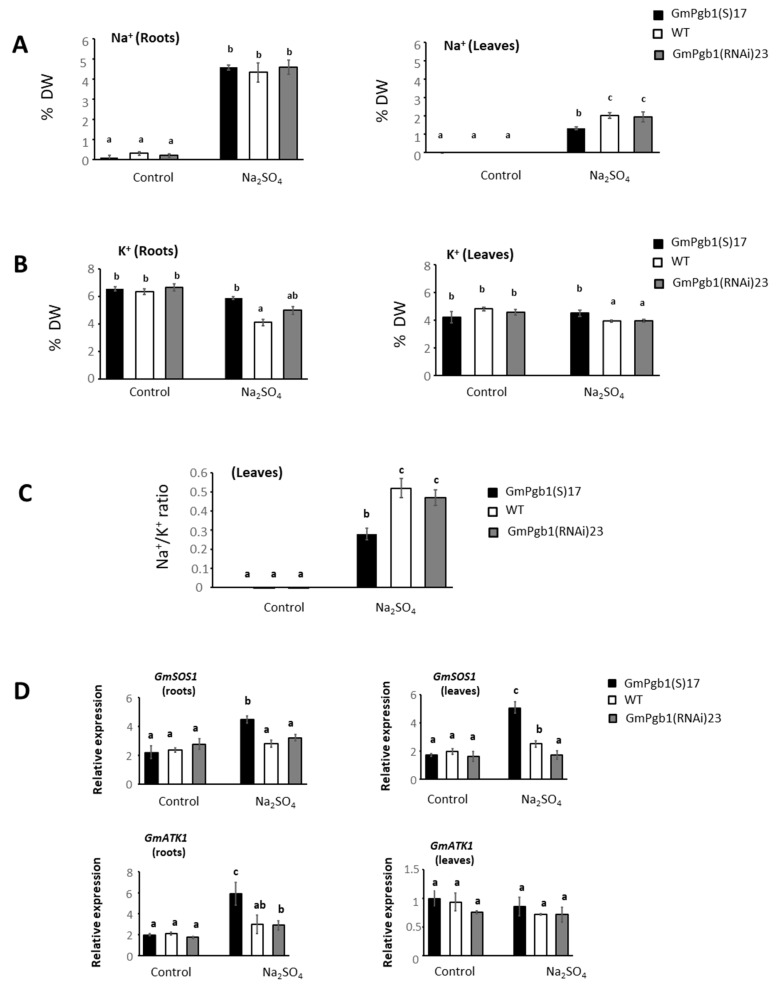
Levels of Na^+^ (**A**), K^+^ (**B**), and Na^+^/K^+^ ratio (**C**) in WT plants and plants overexpressing (GmPgb1(S)17) or downregulating (GmPgb1(RNAi)23) *GmPgb1*. Values are the means ± SE of three biological replicates, each consisting of at least five plants. Letters indicate statistically significant differences (*p* < 0.05). (**D**) Relative transcript levels of *GmSOS1* and *GmAKT1* in the same plants.

**Figure 5 ijms-23-04072-f005:**
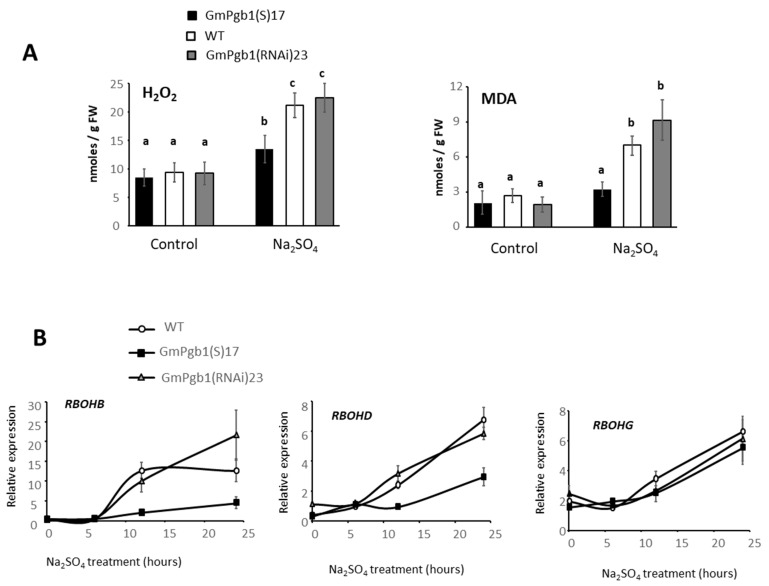
ROS and ROS-induced damage in leaves of soybean plants exposed to Na_2_SO_4_. (**A**) Level of H_2_O_2_ and malondialdehyde (MDA) in leaves of WT plants and plants overexpressing (GmPgb1(S)17) or downregulating (GmPgb1(RNAi)23) *GmPgb1*. Values are the means ± SE of three biological replicates, each consisting of at least three plants. Letters indicate statistically significant differences (*p* < 0.05). (**B**) Relative expression levels of respiratory burst oxidase homologs (RBOHB, D and G) in leaves of plants indicated in (**A**). Values are means ± SE of three biological replicates, each consisting of at least three plants.

**Figure 6 ijms-23-04072-f006:**
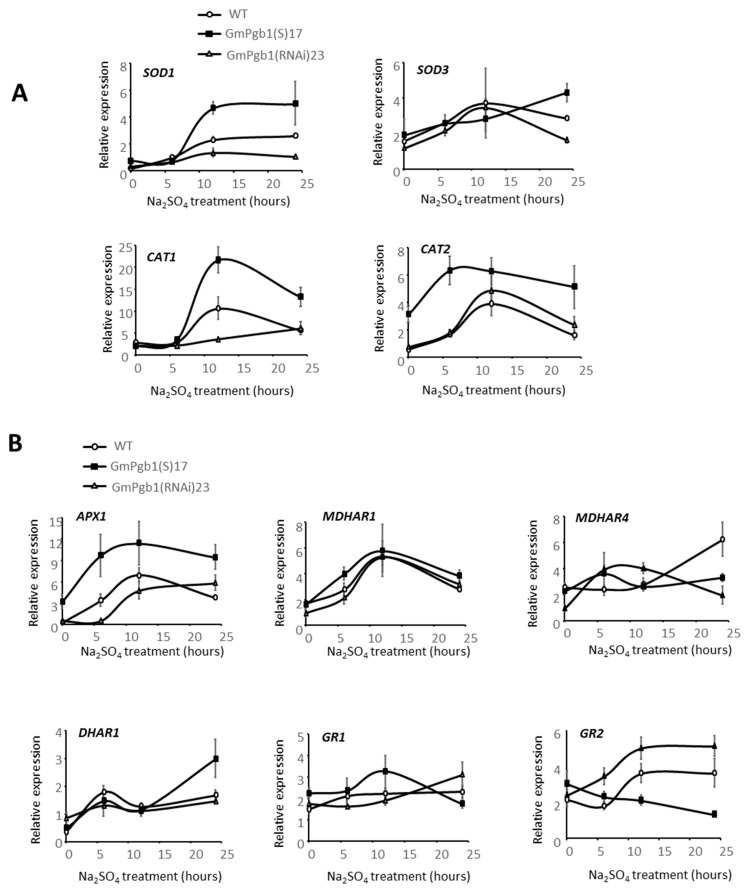
Relative transcript levels of antioxidant enzymes in leaves of soybean plants exposed to Na_2_SO_4_. Measurements were taken in the WT line and the lines overexpressing (GmPgb1(S)17) or downregulating (GmPgb1(RNAi)23) *GmPgb1*. Values are the means ± SE of three biological replicates, each consisting of at least three plants. (**A**) Superoxide dismutase (SOD1, 3) and catalase (CAT 1, 2). (**B**) Ascorbate peroxidase 1 (*APX1*), monodehydroascorbate reductase (MDHAR1, 4), dehydroascorbate reductase (DHAR1), and glutathione reductase (GR1, 2).

**Figure 7 ijms-23-04072-f007:**
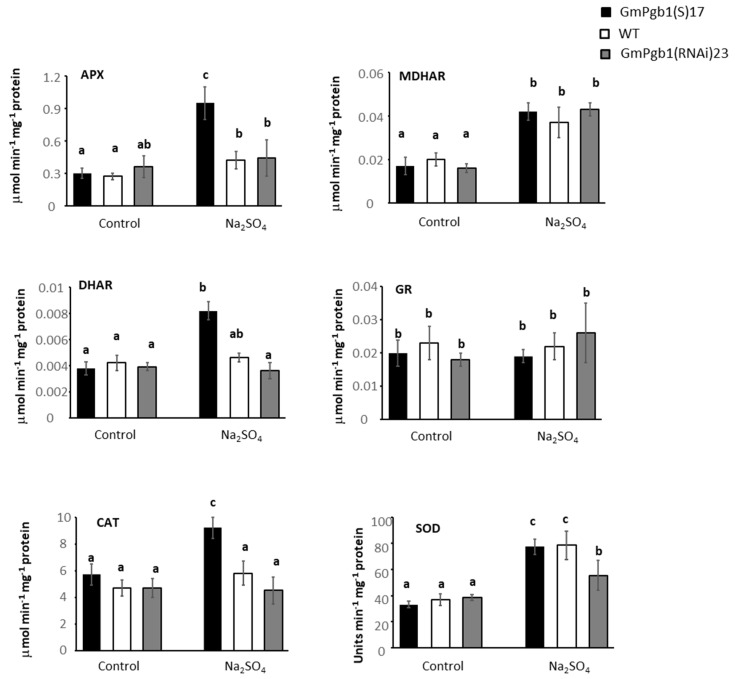
Enzymatic activity of ascorbate peroxidase (APX), monodehydroascorbate reductase (MDHAR), dehydroascorbate reductase (DHAR), glutathione reductase (GR), catalase (CAT), and superoxide dismutase (SOD) in soybean plants exposed to Na_2_SO_4_. Measurements were taken in the WT line and the lines overexpressing (GmPgb1(S)17) or downregulating (GmPgb1(RNAi)23) *GmPgb1*. Values are the means ± SE of three biological replicates, each consisting of at least three plants. Letters indicate statistically significant differences (*p* < 0.05).

**Figure 8 ijms-23-04072-f008:**
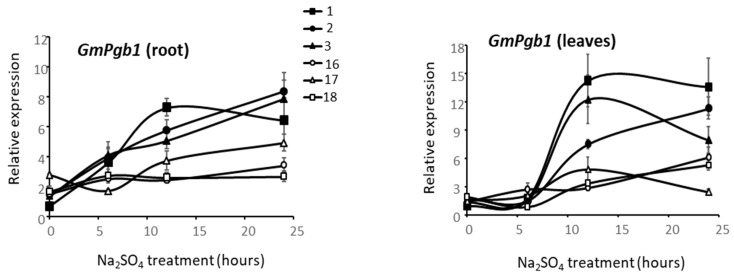
Expression level of *GmPgb1* in roots and leaves of commercial soybean cultivars exposed to Na_2_SO_4_. Cultivars were denoted as tolerant (1, 2, 3) or susceptible (16, 17, 18) to Na_2_SO_4_ stress. Values are the means ± SE of three biological replicates, each consisting of at least three plants.

## Data Availability

The data that support the findings of this study are available from the corresponding author upon reasonable request.

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
