# Peer review of "Phytoglobin Expression Alters the Na+/K+ Balance and Antioxidant Responses in Soybean Plants Exposed to Na2SO4"

_ijms, 2022, doi:10.3390/ijms23084072_

Round 1

Reviewer 1 Report

This manuscript by Youssef et al. investigates the importance of a phytoglobin coded by the GmPgb1 gene on salt tolerance in Glycine max.  The work has been done well from a technical perspective, and the results are clear, compelling, interesting, and important.

This said, I feel the current manuscript does not do justice to the quality of the work, and I think there is substantial work to be done on the structure of the manuscript which will make the final paper more useful to readers.

Broadly, the issues stem from two main sources; 1) the Intro - Results - Discussion - M+M format of this journal, and 2) a lack of clear research goals specified in the final paragraph of the introduction.

1) Paper format
In traditionally formatted journals, authors generally give an overview of the experiment in the first paragraph of the M+M.  However, in these newer format journals, the M+M is buried at the end of the manuscript.  This requires the authors to move that necessary overview to the final paragraph of the Introduction, so that readers can understand what results are going to appear in the next section, and the implications of those results for the manuscript.  This is vitally important.  You need a couple of sentences which begin something like;

"Previous studies have suggested that phytoglobin proteins may have a protective role in plants undergoing abiotic stresses, due to their ability to attenuate oxidative stress.  To investigate this possibility, we grew wild type (WT), GmPgb1 over-expressing (GmPgb1(S)17), or knockdown (GmPgb1(RNAi)23) plants under control conditions, or in the presence of 100mM Na2SO4, and measured ..." (outline briefly the meaning of each measurement)

You can format this however you like, but this type of paragraph will HUGELY help the reader to make sense of what comes next.

2) Hypotheses - or lack thereof
I find your Discussion to be rather unfocused.  Probably, the reason for this is that you lack well-defined hypotheses at the end of your Intro.  Some well specified hypotheses (e.g.  "We hypothesize that plant physiological performance (photosynthetic rates / growth rates) will be higher in GmPgb1 overexpressing plants, and decreased in knockdown plants") will help you refocus your Discussion.  At the moment, I feel you have a bunch of measurements, and you're discussing each one separately, and not showing how they gel together into coherent phenomena.  For example, tieing together the H2O2 data with the SOD transcript and enzyme activity data will better help the reader understand the role of oxidative stress in your system.  At the moment, they don't feel integrated enough to make a coherent story (at least for me).  You have clearly made some attempt to group these together, but I think some well-developed hypotheses will help you group together your measurements into more easily-understandable groups.

Speaking of your Discussion, I thought it a little odd that it took 3 paragraphs to get to GmPGb1.  I would think that should be in the first couple of sentences of the Discussion, and certainly in the first paragraph.  A clear statement that Pgb1 overexpression led to a decrease in salt-induced damage would be important and powerful.

One final point about your writing style which I think you should consider is your paragraph structure.  You have a tendancy to go "context context context, references to other people's work, context, Our work suggests..."

I suggest you lead the paragraph with your observations and their importance, saving the contextualizing and references to previous studies to the end of the paragraph.  I want to know the purpose of a paragraph within the first sentence.  I don't want to have to read half a paragraph before you mention your data.  I'm here for your data!

I want to see your data driving the Discussion more.

One point about data
In Fig 3, I'm not sure it is good practice to present both FW and DW data.  If you think there will be a difference in plant water content, calculate that and present it (I think there is). 

Generally, I would appreciate your figures using common y-axis scales where it is possible.  For example, in Fig 4A, your salt conc. reaches about 5% in roots, and about 2% in leaves, but the bars look to be about the same height, because of the different scales.  Use of a common scale would make it easier for readers to get a more intuitive feel for the data and how the concentrations in the organs relate to each other.

I may also suggest the authors to reconsider the symbols on their diagrams - it can be hard to tell filled-circles and filled-squares apart in small diagrams, but it's vital to be clear.

I also wondered for the data analysis how you dealt with inequality of variance in your data (e.g. your [Na+] data probably fails Levene's test) - probably log transforming or similar, but it isn't mentioned.

There are lots of minor corrections (e.g. lack of subscript on H2O2 in the abstract), but I'll leave those for the next revision.

Your science is interesting and great.  I am sure that we can improve the manuscript to match.

Author Response

This manuscript by Youssef et al. investigates the importance of a phytoglobin coded by the GmPgb1 gene on salt tolerance in Glycine max.  The work has been done well from a technical perspective, and the results are clear, compelling, interesting, and important.

This said, I feel the current manuscript does not do justice to the quality of the work, and I think there is substantial work to be done on the structure of the manuscript which will make the final paper more useful to readers.

Broadly, the issues stem from two main sources; 1) the Intro - Results - Discussion - M+M format of this journal, and 2) a lack of clear research goals specified in the final paragraph of the introduction.

1) Paper format
In traditionally formatted journals, authors generally give an overview of the experiment in the first paragraph of the M+M.  However, in these newer format journals, the M+M is buried at the end of the manuscript.  This requires the authors to move that necessary overview to the final paragraph of the Introduction, so that readers can understand what results are going to appear in the next section, and the implications of those results for the manuscript.  This is vitally important.  You need a couple of sentences which begin something like;

"Previous studies have suggested that phytoglobin proteins may have a protective role in plants undergoing abiotic stresses, due to their ability to attenuate oxidative stress.  To investigate this possibility, we grew wild type (WT), GmPgb1 over-expressing (GmPgb1(S)17), or knockdown (GmPgb1(RNAi)23) plants under control conditions, or in the presence of 100mM Na2SO4, and measured ..." (outline briefly the meaning of each measurement)

You can format this however you like, but this type of paragraph will HUGELY help the reader to make sense of what comes next.

2) Hypotheses - or lack thereof
I find your Discussion to be rather unfocused.  Probably, the reason for this is that you lack well-defined hypotheses at the end of your Intro.  Some well specified hypotheses (e.g.  "We hypothesize that plant physiological performance (photosynthetic rates / growth rates) will be higher in GmPgb1 overexpressing plants, and decreased in knockdown plants") will help you refocus your Discussion.  At the moment, I feel you have a bunch of measurements, and you're discussing each one separately, and not showing how they gel together into coherent phenomena.  For example, tieing together the H2O2 data with the SOD transcript and enzyme activity data will better help the reader understand the role of oxidative stress in your system.  At the moment, they don't feel integrated enough to make a coherent story (at least for me).  You have clearly made some attempt to group these together, but I think some well-developed hypotheses will help you group together your measurements into more easily-understandable groups.

Speaking of your Discussion, I thought it a little odd that it took 3 paragraphs to get to GmPGb1.  I would think that should be in the first couple of sentences of the Discussion, and certainly in the first paragraph.  A clear statement that Pgb1 overexpression led to a decrease in salt-induced damage would be important and powerful.

One final point about your writing style which I think you should consider is your paragraph structure.  You have a tendancy to go "context context context, references to other people's work, context, Our work suggests..."

I suggest you lead the paragraph with your observations and their importance, saving the contextualizing and references to previous studies to the end of the paragraph.  I want to know the purpose of a paragraph within the first sentence.  I don't want to have to read half a paragraph before you mention your data.  I'm here for your data!

I want to see your data driving the Discussion more.

One point about data
In Fig 3, I'm not sure it is good practice to present both FW and DW data.  If you think there will be a difference in plant water content, calculate that and present it (I think there is). 

Generally, I would appreciate your figures using common y-axis scales where it is possible.  For example, in Fig 4A, your salt conc. reaches about 5% in roots, and about 2% in leaves, but the bars look to be about the same height, because of the different scales.  Use of a common scale would make it easier for readers to get a more intuitive feel for the data and how the concentrations in the organs relate to each other.

I may also suggest the authors to reconsider the symbols on their diagrams - it can be hard to tell filled-circles and filled-squares apart in small diagrams, but it's vital to be clear.

I also wondered for the data analysis how you dealt with inequality of variance in your data (e.g. your [Na+] data probably fails Levene's test) - probably log transforming or similar, but it isn't mentioned.

There are lots of minor corrections (e.g. lack of subscript on H2O2 in the abstract), but I'll leave those for the next revision.

OUR RESPONSES

We appreciate the comments of the reviewer that we have addressed in the revised version of the manuscript.  Also, at the end of the introduction we have clearly stated the hypothesis of the work and also indicated that the lines used in this work were developed previously. 

The discussion has also been revised.  The discussion opens with a clear statement describing the function of Pgb1 in attenuating salt-induced damage. Also, following the reviewer’s suggestions we have linked more ROS with antioxidants, and also restructured some of the sentences to present our results first, before citing previous work.  We hope that with these changes the “take home” message is more explicit and direct.

Also, the Y axis of some of the Figures (see Fig. 4A) has been changed to better appreciate differences in Na+ and K+  content among organs, and the symbols of the line graphs have been changed.  The marker of the down-regulating line has now been filled with grey.  The graphs look very clear from our side, hopefully their dimension will not be reduced too much when formatted by the journal’s office.  Finally, we have calculated the water content.

Reviewer 2 Report

This work can be accepted for publication after the authors make some clarifications.
1. you must specify the name of the Measurements of K+ and Na+ method, and not just the laboratory that determined it.
2. DAF-FM DA in DMSO, was there a solvent control
3. At a wavelength of 460, the leaf can give autofluorescence, how did the authors distinguish the true signal from it?
4. Why do the authors not provide a photo of visualization of the dye signal? It would be clearly understandable how the untreated control differs from the experiment, and whether the dye on the whole sheet really gives the true signal. The lack of photos and visualization of the controls makes me question the validity of the data. 

Author Response

This work can be accepted for publication after the authors make some clarifications.
1. you must specify the name of the Measurements of K+ and Na+ method, and not just the laboratory that determined it.
2. DAF-FM DA in DMSO, was there a solvent control
3. At a wavelength of 460, the leaf can give autofluorescence, how did the authors distinguish the true signal from it?
4. Why do the authors not provide a photo of visualization of the dye signal? It would be clearly understandable how the untreated control differs from the experiment, and whether the dye on the whole sheet really gives the true signal. The lack of photos and visualization of the controls makes me question the validity of the data. 

All the pictures shown are “normalized” to the DMSO control.  That is, before visualizing the images, the microscope setting was adjusted so that the autofluorescence in the DMSO control pics disappeared (the DMSO control images are therefore completely black).  With the adjustments, the visible signal is only the result of the DAF-FM DA signal.  If needed, we can provide the DMSO-control pics, which are just black.

Round 2

Reviewer 1 Report

Dear Authors,

I am glad to see a strong improvement in the manuscript, especially the Discussion, which is hugely stronger.

I would still like to see more development of the hypotheses at the end of the introduction -- it's really not clear what types of measurements you're going to present, and why you chose those measurements (although you do some of this work at the start of each results subsection).

I would suggest that you outline that you're going to measure several physiological parameters (biomass, water content, leaf damage, K/Na concentrations), photosynthetic parameters (gas exchange and stomatal measurements) and measurements related to oxidative stress (enzyme activities, lipid peroxidation, etc).  I think explaining the relevance of each of these clusters of measurements will help readers grasp the bigger picture.

For the Discussion, I suggest you tone down the certainty of the language a little.  You have a tendency to be overly-confident in your declarations of why things are happening, e.g.
"These are the consequence of the combinatorial effects of salinity resulting from osmotic stress reducing the plant's hydraulic conductivity, anf most importantly ionic stress caused my the toxic accumulation of ions, especially Na+"

I think it's OK to talk about your results as being "consistent with" or pointing towards this explanation, but I think it's best to use more nuanced language in your explanations.

In your Intro, near the end you talk about pharmacological applications of NO.  I couldn't really understand what this paragraph was trying to convery, so i suggest you perhaps look at rephrasing it.

Finally, there are many sentences where the phrasing is a little odd (e.g. first line of Intro!), or could be tightened up (e.g. several conditions of stress --> several stress conditions;  acting as major antioxidant --> acting as the major antioxidant(s?)).   I suggest checking the manuscript carefully (or asking a native-speaking colleague to check) to ensure readability.

Author Response

  • I would still like to see more development of the hypotheses at the end of the introduction -- it's really not clear what types of measurements you're going to present, and why you chose those measurements (although you do some of this work at the start of each results subsection). I would suggest that you outline that you're going to measure several physiological parameters (biomass, water content, leaf damage, K/Na concentrations), photosynthetic parameters (gas exchange and stomatal measurements) and measurements related to oxidative stress (enzyme activities, lipid peroxidation, etc).  I think explaining the relevance of each of these clusters of measurements will help readers grasp the bigger picture.

We have included the suggested section at the end of the introduction

  • For the Discussion, I suggest you tone down the certainty of the language a little.  You have a tendency to be overly-confident in your declarations of why things are happening, e.g.
    "These are the consequence of the combinatorial effects of salinity resulting from osmotic stress reducing the plant's hydraulic conductivity, anf most importantly ionic stress caused my the toxic accumulation of ions, especially Na+" I think it's OK to talk about your results as being "consistent with" or pointing towards this explanation, but I think it's best to use more nuanced language in your explanations.

We have addressed this comment in several instances

  • In your Intro, near the end you talk about pharmacological applications of NO.  I couldn't really understand what this paragraph was trying to convey, so i suggest you perhaps look at rephrasing it.

We have now rephrased the sentence with the purpose to provide evidence that a transgenic approach is preferable to pharmacological treatments when manipulating NO

  • Finally, there are many sentences where the phrasing is a little odd (e.g. first line of Intro!), or could be tightened up (e.g. several conditions of stress --> several stress conditions;  acting as major antioxidant --> acting as the major antioxidant(s?)).   I suggest checking the manuscript carefully (or asking a native-speaking colleague to check) to ensure readability.

We have corrected several sentences, as suggested

Reviewer 2 Report

Dear authors, you need to provide not only control photos. I strongly recommend that you supplement the article with a drawing with images of control and prototype samples. It will certainly enhance the article. It is also completely unclear to me why you did not use the Microsoft Word template (https://www.mdpi.com/files/word-templates/ijms-template.dot) or LaTeX template to create the manuscript. Please prepare the manuscript according to all the rules.

Author Response

  • you need to provide not only control photos. I strongly recommend that you supplement the article with a drawing with images of control and prototype samples. It will certainly enhance the article.

As indicated in our previous response to the reviewer, control images are just black micrographs, as the setting of the microscope eliminated autofluorescence (as stated in the figure legend). This approach ensures that the visible signal is exclusively the result of the treatment.  However, we have now included a control (unstained) micrographs, which we feel it is unnecessary. 

  • It is also completely unclear to me why you did not use the Microsoft Word template (https://www.mdpi.com/files/word-templates/ijms-template.dot) or LaTeX template to create the manuscript. Please prepare the manuscript according to all the rules.

The submitted revision uses the journal template

Round 3

Reviewer 1 Report

The manuscript is generally improved.  However, there are many grammatical errors and poorly phrased sentences.

The authors did not respond to my request to explain about how they dealt with data which may fail Levene's test.  I expect this to be remedied in the final version.

My main issue is with the figure text.  Most of the axes are almost unreadable, as are the letters denoting significance (e.g. in Fig 4d).  You should also remove the letter "a" floating in the space above Fig 3 (root DW).

Minor corrections

L18; exhibited --> resulting in

L30; plant's --> plant

L32; "and the survival of the plant will depend on the..." --> with plant survival dependent on the...."

L41; take water -->absorb water

L46; because of --> as a result of

L52;  don't understand what you mean by "water management"

L80; "dealing with the stress" --> stress tolerance

L108; transcripts --> transcript

L124; 24 of --> 24h of

L125; water used efficiency --> water use efficiency

L129; the closure of the stomata --> stomatal closure

L130; "suggested" --> "evidenced" of "which manifested as..."

L147; Morphological changes - do you mean morphological differences?

L237; "many enzymes" - unnecessarily vague

L265; alleviates --> alleviated

L269; water use efficiency

L269: suggest "can be" --> may be

L286;  "and, as well" -- doesn't make sense.

L289;  the increased in SOS --> increase

L298; salt stressed

L302; delete "unequivocally"

L324; the CAT --> CAT

I suggest one of the native English speaking co-authors carefully checks the manuscript before the final version is uploaded.

Author Response

Please see the attach.

Reviewer 2 Report

Please remove "Fig.4" from the figure itself. And I would like a denser arrangement of figures and text, so that there is not half a blank page.

Author Response

Please see the attach.
